# Tunnelled Haemodialysis Catheter Removal: An Underappreciated Problem, Not Always Simple and Safe

**DOI:** 10.3390/ijerph17093027

**Published:** 2020-04-27

**Authors:** Tomasz Porazko, Jacek Hobot, Zbigniew Ziembik, Marian Klinger

**Affiliations:** 1Department of Nephrology and Internal Medicine, Institute of Medical Sciences, University of Opole, 45-052 Opole, Poland; marian.klinger@uni.opole.pl; 2Department of General and Vascular Surgery, Institute of Medicine, University of Opole, 45-052 Opole, Poland; jhobot@uni.opole.pl; 3Institute of Biotechnology, University of Opole, 45-052 Opole, Poland; ziembik@uni.opole.pl

**Keywords:** chronic kidneys disease, hemodialysis, vascular access, central tunneled catheters

## Abstract

Background: Optimal care of patients treated with a central tunneled catheter (CTC) as vascular access for hemodialysis requires a number of procedures. One of them is CTC removal, usually carried out using mostly the cut-down method (CDM) and the traction method (TM). The procedure seems to be simple and safe; however, occasionally, serious complications may occur. To eliminate the risk of such events, we have introduced a modified cut-down method (MCDM). Methods: The study included the analysis of retrospective results of 143 CTC removal procedures, 76 of which were performed using the standard cut-down method (CDM), and in 67 cases, the modified cut-down method (MCDM) was applied. Results: As minor side effects occurred in patients treated with both methods with comparable frequency, serious complications were observed only in the CDM patients group. Conclusions: In our opinion, the new MCDM procedure is the simplest and safest method of CTC removal.

## 1. Introduction

Central tunneled catheters (CTCs) are widely used as vascular access for hemodialysis in a significant number of prevailed end-stage renal disease (ESRD) patients [1]. The main attention is focused on the improvement of the line insertion technique and everyday care to maintain a high level of CTC patency without thrombotic dysfunction or infective complications. In comparison with that, a negligible interest is paid to CTC removal, which is believed to be a simple and safe even in the outpatient setting [2,3]. However, there are some significant risks that cannot be ignored [2,3,4]. There are two main indications for the operation. The first one is a scheduled removal, when a CTC is not needed anymore, i.e., for a patient with matured arteriovenous fistula (AVF) or graft (AVG), as well as with a successfully functioning kidney transplant. The second indication results from a medical emergency due to unexpected complications, like catheter-related infection (CRI) or thrombotic dysfunction requiring an exchange. Generally, two techniques of CTC removal are widely used: the cut-down method (CDM) [5,6,7] and the traction method (TM) [2,3], or each one with a few modifications [7,8]. Despite their simplicity and safety, both methods can be seriously complicated. In the case of technique failures of CDM, there were reports of an intravascular part of CTC migrating into superior vena cava (SVC) or right atrium (RA) [4]. CDM was regularly used both in our center and in its satellite units from 2008 till 2015. However, after the observation of two patients with a CTC part-migration and one patient with a stuck CTC, our own modification of catheter removal was devised in 2015, and it has been used ever since. In principle, in the modified approach, after the CTC cuff is released, the intravascular part is pulled out and cut down just after, instead of before, as it was done previously in the CDM method. We believed (hypothesized) that this would prevent the inadvertent complications of migration of the distal part of CTC into SVC or RA or retained segment, as previously observed in our practice and also reported by others.

We report our retrospective analysis of our experience in the application of the new variant modified cut-down technique (MCDM) compared to the standard CDM technique.

## 2. Materials and Methods

The study was a retrospective analysis of the available, complete patient records of the Department of Nephrology and Dialysis Unit and the Department of General and Vascular Surgery of Opole University Hospital from August 2008 to December 2018. From 2008 to 2018, 172 patients had CTCs removed. The MCDM has been introduced in our center since 2015. Further analysis included complete records of 143 patients. Two methods were mainly used: CDM in 76 patients, and MCDM in 67 patients. There was only a small group of 17 patients who had the CTC removed with TM, and it was not taken into analysis.

MCDM is a modification of CDM described previously [5,6,7]. Simply, after a CTC cuff localization, using sterile technique and Trendelenburg position, local anesthetic (i.e., 1% lignocaine) is injected at an area, and afterwards, a 2-cm incision was made along the cuff. Using a dissector, surrounding tissues were removed and the cuff is freed (Figure 1). Then, without cutting the CTC line, the intravenous part is pulled out from the SVC through the skin incision, with pressure applied at the vein entering point for a few minutes (Figure 2). Following that, the CTC is cut distally to a cuff, and the distal part of the CTC is removed through the exit site (Figure 3). Hemostasis is achieved, and the skin incision is closed with 2 to 3 non-absorbable 3.0 sutures, and a dressing is applied. Contrary to MCDM, in CDM, as soon as the cuff is exposed, the CTC line is clamped and then cut before its intravenous part is removed from the vein, with or without extra sutures, as described previously [7].

We used TM exclusively in our patients when the CTC cuff was not yet grown-in or dislocated. The minority of all patients were referred for CTC removal. In short, under sterile conditions, after the tunnel from the exit site to the cuff position is anesthetized, the CTC is removed by simply pulling. When necessary, wound margins are properly prepared; it is closed with non-absorbable suture, and a dressing is applied. In the cases when the cuff is disconnected to the CTC line, the CTC is pulled out, and the cuff is removed through an incision like in CDM and MCDM. All procedures were carried out by nephrologists in our department. If serious complications occurred after CTC removal, the patients were treated by vascular surgeons.

Patients data were analyzed according to demographic characteristics (i.e., gender, age, cause of end—stage kidney disease (ESKD), CTC site (i.e., right internal jugular vein (RIJV), left internal jugular vein (LIJV)), the time elapsed between CTC insertion and its removal, indication for the procedure and the complications which occurred within following 10 days (i.e., infectious complications, bleeding from post-operative wounds and procedure-specific complications like air embolism, CTC line defragmentation, cuff retention) till the removal of the sutures.

All the procedures under study were aimed at restoring vascular access function for hemodialysis and life support therapy. These procedures were conducted in accordance with the ethical standards of the Declaration of Helsinki from 1964 and its later amendments. Informed consent was obtained from all the individual patients before every procedure. The patients presented in the images illustrating MCDM have given their written consent for publishing the materials.

### Statistical Analysis

The analysis was conducted with the use of statistical software R (version 3.5.2; http://cran.r-project.org, The R Foundation for Statistical Computing, Vienna, Austria). The normality of distribution was verified on the basis of the Shapiro–Wilk test, the kurtosis and skewness values, as well after the visual assessment of histograms. The data are presented as absolute frequencies (n) and % of the group for nominal variables and as mean (SD) or median and interquartile range (IQR: Q1; Q3) for continuous variables. The comparison of the groups was conducted with the Fisher exact test, *χ*^2^ test, independent samples Student *t*-test or the Mann–Whitney U-test, as appropriate. Additionally, relative risk (RR) between the groups analyzed with 95% confidence interval (CI) was calculated for complications. All tests were two-tailed, and differences were considered significant at the level of *p* < 0.05.

## 3. Results

Among all the patients in whom a CTC removal was performed, there were 3 cases of serious complications treated in the departments of Opole University Hospital, which motivated us to change the procedure.

Case 1: a 37-year-old man was admitted to hospital for the CTC removal procedure as he had been dialyzed for a few weeks, effectively using the radio-cephalic AVF. After standard preparation, under local anesthesia, using the cut-down method, a 2-cm incision was made along the CTC cuff. After the dissection of the surrounding tissues, the cuff was exposed; then, the CTC above the cuff was clamped. Due to a mistake, the CTC was cut above the clamp instead of between the clamp and the cuff, and the proximal part of the CTC migrated to RIJV. An urgent supine chest and abdomen X- ray was performed, showing the intravascular part of the CTC located in the junction between the superior vena cava (SVC) and the inferior vena cava (IVC) (Figure 4). The patient was given 1.0 g of vancomycin as an antibiotic prophylaxis and transferred to Interventional Cardiology Department procedure room, where, with the assistance of a vascular surgeon, the interventional cardiologist, under general anesthesia, introduced a snare catheter (Andra Snare, Andramed, Reutlingen, Germany) through the right femoral vein (RFV). The intravascular CTC part was transferred to the RFV. An open venotomy was performed and the CTC part was removed, the vein was closed with vascular suture, and the wound was closed. The patient recovered well with no further major complications.

Case 2: a 47-year-old woman was transferred to the Department of Vascular Surgery at Opole University Hospital from a district hospital, with the proximal part of the CTC having migrated to SVC, damaged during the removal procedure, using tracking methods previously described [2,3]. The indication was the CTC dysfunction. It turned out that under local anesthesia, when pulling off the CTC was ineffective, using 20 cm hemostat, a blind dissection through the exit site was followed by pulling the CTC line, which resulted in a sudden CTC line defragmentation proximally to the cuff, with the intravascular CTC part migrating into SVC, and the distal part of the CTC with the cuff was pulled out. The chest X-ray showed the CTC part in the junction of RIJV and SVC (Figure 5). Before the procedure, the patient was given 1.0 g vancomycin as a prophylaxis. Under general anesthesia, a 5-cm skin incision was made along the lateral aspect of the right sternocleidomastoid muscle, and after a blind dissection, the right internal jugular vein was exposed. A 3-cm venotomy was made, and a 15-cm intravascular part of the CTC was removed. The vein was closed with vascular suture, a drain was inserted locally, and the wound was closed with suture. After two days, the drain was removed. The patient recovered well, dialyzed using a temporal CVC inserted into the right femoral vein. After a week, a new CTC was inserted into the left internal jugular vein.

Case 3: a 52-year-old woman was admitted to the Department of Nephrology of Opole University Hospital due to a CTC dysfunction over guidewire exchange. Using CDM, the CTC cuff was exposed, the line was clamped proximally, and the distal part was cut and removed through the exit site. In an attempt to remove the intravascular part of CTC, the patient started to complain about the pain, which was located in the mid-sternum area, and resistance was felt. As the procedure was carried out in the interventional room of the dialysis unit, without access to fluoroscopy, the CTC part was double-clamped, secured with an additional suture applied to the skin, and the patient was transferred to endovascular suite. Endoluminal dilatation was made a few times, as described previously [9,10], over guidewire passed through both lumens of the CTC, using two standard balloons (Armada 4/60 mm, Abbott, Abbott Park, USA and Passeo 4/60 mm, Biotronic, Berlin, Germany). However, it was ineffective, and finally, the proximal part of the CTC was successfully removed by means of a snare (Andra Snare, Andramed, Reutlingen, Germany) using the method described elsewhere (Figure 6). A venography of the RIJV, as well as the SVC, did not show any complications. A new CTC was implanted into the RIJV. The CTC function was checked and hemodialysis performed without complications. The patient was discharged the next day.

As mentioned before, a group of 172 patients underwent CTC removal in our center between 2011 and 2019. The complete, available records of 143 patients were analyzed further. MCDM was used in 67 patients and CDM in 76 patients. The summary of the groups’ demographic and clinical characteristics is presented in Table 1. There were no statistical differences between MCDM and CDM in gender (53 (9.1%) females and 14 (20.9%) males vs. 49 (64.5%) females and 27 (35.5%) males, respectively) and age (mean ± SD, 65.72 ± 16.04 vs. 63.16 ± 13.33 years, respectively). Diabetes mellitus (DM) was the leading cause of ESRD (29 (43.3%) pts vs. 37 (48.7%) pts), followed by neoplasm (NPL); 15 (22.4%) pts vs. 11 (14.5%) pts) and glomerulonephritis (GN; 12 (17.9%) pts vs 8 (10.5%) pts), of comparable occurrence in the MCDM and the CDM group, respectively. Time of CTC removal after insertion did not differ between the MCDM and the CDM patients (median 57.00 (42.00; 71.50) weeks vs. 59.00 (36.75; 87.00) weeks, respectively). Catheters were mostly removed using MCDM and CDM from the right (44 (65.7%) vs. 49 (64.5%) pts, respectively) and from the left (14 (20.9%) pts vs. 23 (30.3%) pts, respectively) jugular vein. The main indication for the procedure in both, the MCDM and the CDM group, were patent AVF or AVG (33 (34.3%) cases vs. 40 (52.6%) cases, respectively) followed by CTC dysfunction (31 (46.3%) cases vs. 24 (31.6%) cases).

Postprocedural complications occurred with comparable frequency in 9 (13.4%) MCDM patients and in 11 (14.5%) patients form the CDM group (Table 2). However, prolonged bleeding not requiring blood transfusion was the most frequent (5 (5.5%) cases vs. 6 (7.9%) cases, respectively). All cases were treated with compression and dressing exchange, and additional sutures were added when necessary. A subsequent side effect was a postoperative wound infection (4 (6.0%) cases vs. 3 (3.6%) cases, respectively), mostly treated with a course of an oral antibiotic. However, 2 (2.6%) cases of CTC migration into SVC requiring surgical intervention and 1 (1.3%) case of air embolism ending with a successful medical treatment occurred only in the CDM group. Further analysis revealed that in the MCDM group, patients with bleeding after CTC removal were significantly older than patients without bleeding (MD = 16.5, CI 95 (2.00; 28.00), p = 0.023). No risk factors were confirmed for the presence of infection in the MCDM patients (Table 3 and Table 4). For CDM patients, no significant risk factors were identified for the presence of bleeding or infection after CTC removal (Table 5 and Table 6). Serious complications occurrence was influenced only by the CTC removal method that included the standard cut-down method.

## 4. Discussion

In the presented material, MCDM was applied in 67 patients compared with standard CDM in 76 patients. In the majority of patients, CTC was removed as a scheduled procedure due to the availability of permanent vascular access (88% in MCDM and 79% in CDM). In the remaining cases, CTCs were resected due to thrombotic or infectious complications. It should be emphasized, that serious complications occurred exclusively in the CDM group, i.e., two (2%) described cases of CTC migration into SVC, which inspired the MCDM development. There was also 1 (1%) case of air embolism, followed by a successful medical treatment, in the CDM group. The remaining complications were minor and occurred rarely and were not connected with the cause of removal (scheduled or urgent) and the method used (MCDM vs. CDM). In detail, 5 (7%) cases of local bleeding appeared in MCDM, as compared with 6 (8%) in CDM. Post-procedure wound infection occurred in 4 (5%) and 3 (4%) cases in MCDM and CDM, respectively. In addition, one case of a failed removal due to a stuck CTC was observed. Similar cases were reported in patient groups in whom TM was applied to remove the CTC [3].

The traction method (TM) was not analyzed in this material because of its limited use exclusively in a non-grown-in CTC or in the case of a CTC cuff with infection complicated with a CTC that has fallen out. When applied to a CTC with a well-matured cuff, TM was mainly complicated—according to literature—with cuff retention, with occurrence ranged from 0% to 32% [2,3] depending on CTC type. It may occasionally lead to infection [11], prolonged healing, or misinterpretation of radiograms [12,13]. Post-procedure prolonged bleeding after using TM ranged from 1.4% to 8% [3,14]. We have observed a similar incidence of that complication; all patients required only a dressing exchange and prolonged pressure application. In another report of a small group of 8 patients, in whom a specially designed device—a transcatheter extractor—was used [8]. Patients were advised to remain sitting for up to six hours to avoid bleeding. There were no data presented regarding post-procedure complications. Catheter removal with TM leaves a pocket-like scar of an exit site. When the cuff is close to the exit site, using MCDM fine suturing technique, all scaring tissues can be removed with the CTC, which results in a good cosmetic effect. In our study, using MCDM, we have not observed a CTC defragmentation or migration to the central vein. It may occur when CDM is used (Case 1). Application of TM may cause a CTC rupture, especially in certain types of catheters and in the case of catheters used for more than 3 to 4 years [2,3]. After the implementation of MCDM, we did not observe any such cases, as the CTC is firstly removed from the vein and then cut. In the presented case (Case 2), the traction method was used, as described by Fülöp et al. [3], and the time from insertion to removal was 89 weeks. All cases required sophisticated endovascular and surgical methods (Case 1) [15].

We observed one case of a failed removal due to a stuck CTC (Case 3). It is similar to the reports of a patient group treated with TM [3]. In our opinion, the MCDM application provides an earlier diagnosis of such complications because the only resistance is produced by pathological adhesion between the CTC line and the vein wall, not by the CTC cuff. Therefore, there is a lower risk of a vein wall rupture when MCDM is applied. In comparison with CDM, MCDM also provides an additional advantage. When a stuck CTC in the central vein is diagnosed, while the removal procedure is done at the bedside or in the outpatient setting, the non-cut CTC can be left safely in, the wound can be sutured, and the patient can be safely transferred to the endovascular unit without the need for an extra CTC line securing, as required in CDM [10].

## 5. Conclusions

Central tunneled catheter removal seems to be a safe and simple procedure, even at the bedside of the patient. None of all previously described methods, however, were performed without complication. In our opinion, MCDM offers a safer and simpler alternative way to remove CTCs. The limitations of our observation are a single-center experience, retrospective character, and a small sample of patients. Even so, our data may contribute to the improvement of CTC removal technique. The presented modified cut-down method may increase the safety of a central tunneled catheter removal and protect against nonintentional catheter migration into central veins or enhance stuck catheter removal.

## Figures and Tables

**Figure 1 ijerph-17-03027-f001:**
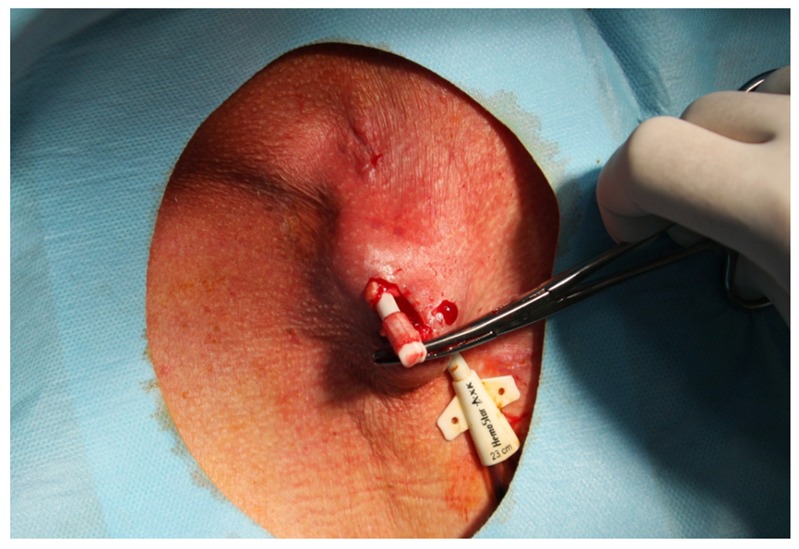
Under local anesthesia, a 2-cm incision is made, and, using blind dissection, the central tunneled catheter (CTC) cuff is exposed and freed from surrounding tissues.

**Figure 2 ijerph-17-03027-f002:**
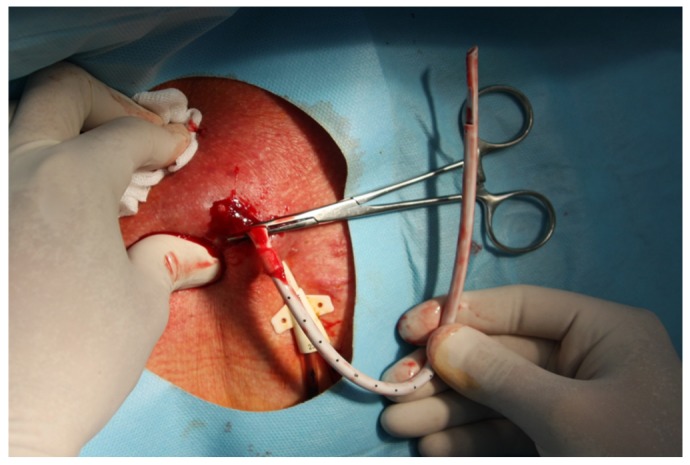
The intravenous part of a CTC is pulled out from a vein without cutting, through a skin incision, simultaneous with pressure applied at the vein entering point for a few minutes.

**Figure 3 ijerph-17-03027-f003:**
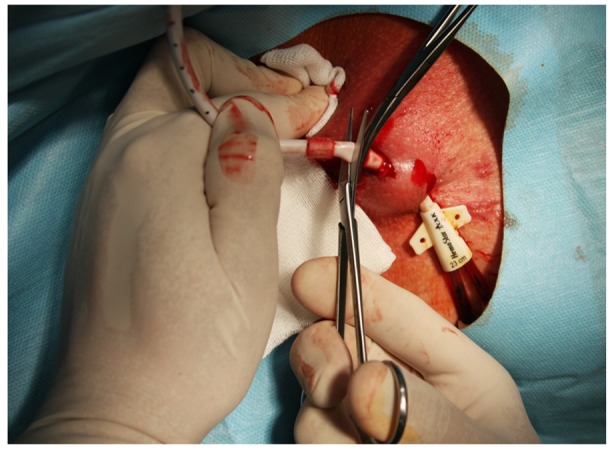
When CTC is pulled out from the vein completely, it is cut distally to a cuff. The distal part of the CTC is removed through the exit site. After hemostasis is achieved, the wound is sutured.

**Figure 4 ijerph-17-03027-f004:**
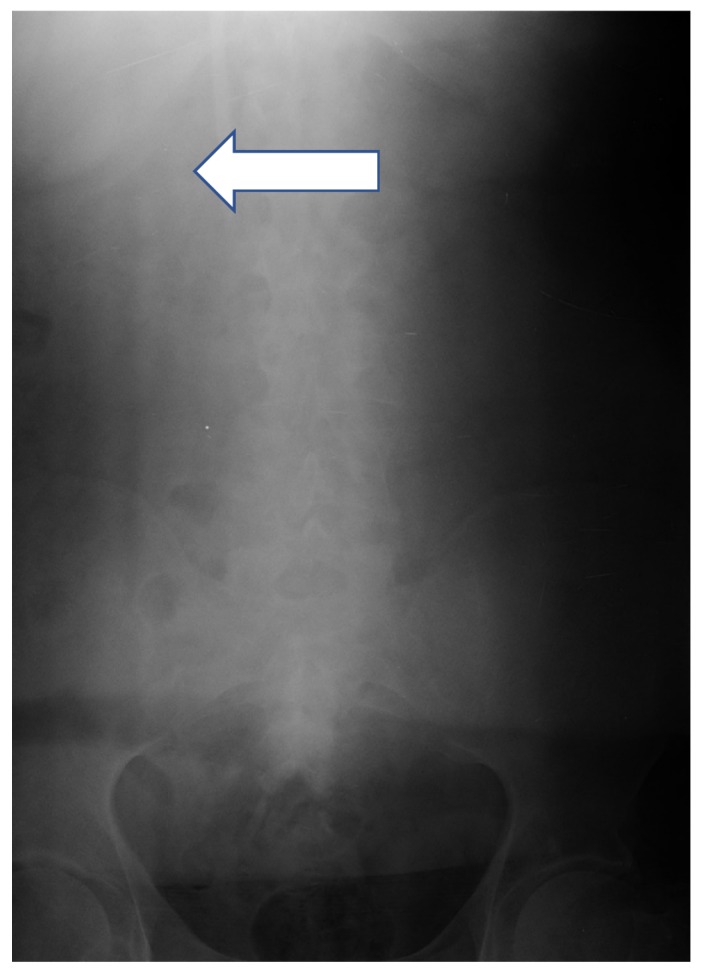
Supine abdominal X-ray showing intravascular part of CTC located in the junction between superior vena cava (SVC) and inferior vena cava (IVC) (arrow).

**Figure 5 ijerph-17-03027-f005:**
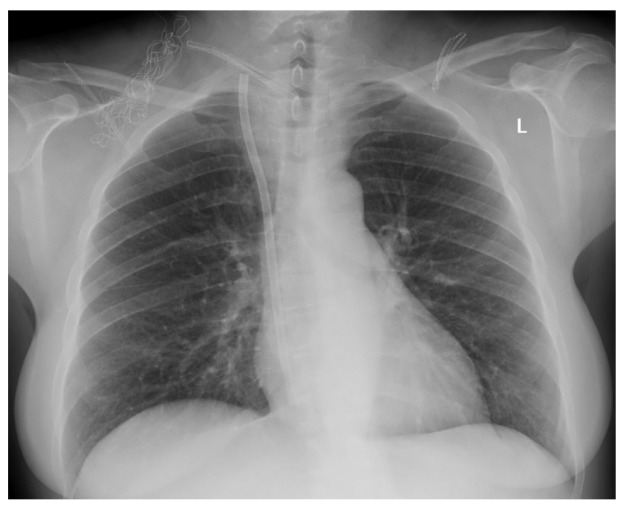
A chest X-ray showed the intravascular part of a CTC in SVC.

**Figure 6 ijerph-17-03027-f006:**
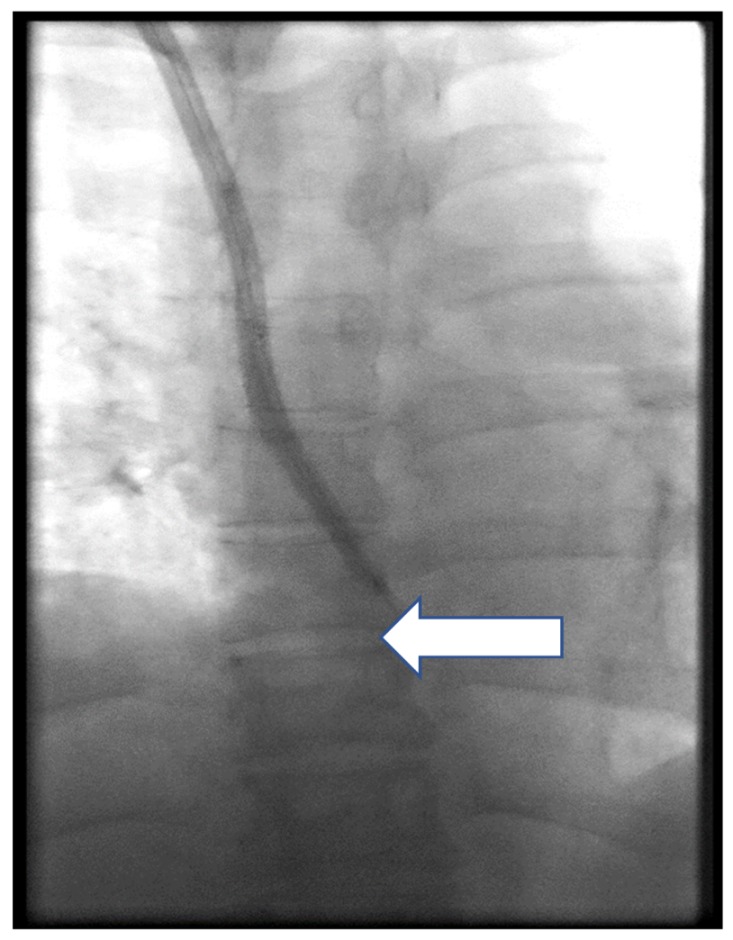
The intravascular part of the CTC was successfully removed using a snare (Andra Snare, Andramed; arrow).

**Table 1 ijerph-17-03027-t001:** Demographic and clinical characteristics of study groups.

Characteristic	MCDM Group	CDM Group	*p*
*N*	67	76	
Sex, *n* (%)			
Female	53 (79.1)	49 (64.5)	0.081
Male	14 (20.9)	27 (35.5)	
Age, years, mean ± *SD*	65.72 ± 16.04	63.16 ± 13.33	0.299
ESRD cause, *n* (%)			
ADKPD	3 (4.5)	7 (9.2)	0.464
DM	29 (43.3)	37 (48.7)	
GN	12 (17.9)	8 (10.5)	
HTN	5 (7.5)	7 (9.2)	
NPL	15 (22.4)	11 (14.5)	
UNKN	3 (4.5)	6 (7.9)	
Reason for removal, *n* (%)			
AVF	21 (31.3)	33 (43.4)	0.118
AVG	2 (3.0)	7 (9.2)	
CRI	13 (19.4)	11 (14.5)	
DYS	31 (46.3)	24 (31.6)	
PD	0 (0.0)	1 (1.3)	
CTC removal site, *n* (%)			
LFV	4 (6.0)	1 (1.3)	0.487
LJV	14 (20.9)	23 (30.3)	
LSV	1 (1.5)	1 (1.3)	
RFV	2 (3.0)	1 (1.3)	
RJV	44 (65.7)	49 (64.5)	
RSV	2 (3.0)	1 (1.3)	
CTC time at removal, weeks, median (Q1; Q3)	57.00 (42.00; 71.50)	55.00 (22.25; 94.00)	0.433

Data presented as *n* (% of group) unless otherwise indicated. Groups compared with χ^2^ test or Fisher exact test for nominal variables and *t*-test or Mann–Whitney U-test for continuous variables.CDM: cut-down method, MCDM: modified cut-down method, ESRD: end-stage renal disease, ADPKD: adult polycystic kidney disease, DM: diabetes mellitus, GN: glomerulonephritis, HTN: hypertension, NPL: neoplasm, UNKN: unknown, AVF: matured AV fistula, AVG: matured AV graft, CRI: catheter-related infection, DYS: CTC dysfunction, PD: conversion to peritoneal dialysis, RJV: right jugular vein, LJV: left jugular vein, RSV: right subclavian vein, LSV: left subclavian vein, RFV: right femoral vein, LFV: left femoral vein, CTC: central tunneled catheter.

**Table 2 ijerph-17-03027-t002:** Frequency of CTC removal complications between groups.

CTC Removal Complication	MCDM Group	CDM Group	RR (95% CI)	*p*
n	67	76		
Any complication	9 (13.4)	11 (14.5)	1.08 (0.47; 2.44)	>0.999
Bleeding	5 (7.5)	6 (7.9)	1.06 (0.34; 3.31)	>0.999
Wound infection	4 (6.0)	3 (3.9)	0.66 (0.15; 2.85)	0.706
Air embolism	0 (0.0)	1 (1.3)	n/a	>0.999
CTC migration	0 (0.0)	2 (2.6)	n/a	0.499

Data presented as *n* (% of group). Groups compared with χ^2^ test or Fisher exact test. *RR*—relative risk of complications presence in CDM group vs. MCDM group with 95% confidence interval (CI).

**Table 3 ijerph-17-03027-t003:** Comparison of modified cut-down method (MCDM) patients with and without bleeding complications.

Characteristic	MCDM Group: No Bleeding	MCDM Group: Bleeding	*p*
*N*	62	5	
Sex, *n* (%)			
Female	48 (77.4)	5 (100.0)	0.576
Male	14 (22.6)	0 (0.0)	
Age, years, median (Q1; Q3)	66.50 (57.00; 74.25)	83.00 (73.00; 85.00)	0.023
ESRD cause, *n* (%)			
ADKPD	3 (4.8)	0 (0.0)	0.192
Diabetes	28 (45.2)	1 (20.0)	
Glomerulonephritis	12 (19.4)	0 (0.0)	
Hypertension	4 (6.5)	1 (20.0)	
Neoplasm	12 (19.4)	3 (60.0)	
Unknown	3 (4.8)	0 (0.0)	
Reason for removal, *n* (%)			
Patent AVF	19 (30.6)	2 (40.0)	0.877
Patent AVG	2 (3.2)	0 (0.0)	
Catheter related infection	13 (21.0)	0 (0.0)	
Cathteter dysfunction	28 (45.2)	3 (60.0)	
Conversion to PD	19 (30.6)	2 (40.0)	
CTC removal site, *n* (%)			
Left femoral vein	4 (6.5)	0 (0.0)	>0.999
Left jugular vein	13 (21.0)	1 (20.0)	
Left subclavian vein	1 (1.6)	0 (0.0)	
Right femoral vein	2 (3.2)	0 (0.0)	
Right jugular vein	40 (64.5)	4 (80.0)	
Right subclavian vein	2 (3.2)	0 (0.0)	
CTC time at removal, weeks, median (*Q1*; *Q3*)	57.00 (42.50; 71.75)	37.00 (21.00; 59.00)	0.202

Data presented as *n* (% of group) unless otherwise indicated. Groups compared with Fisher exact test for nominal variables and Mann–Whitney U-test for continuous variables.

**Table 4 ijerph-17-03027-t004:** Comparison of MCDM patients with and without infection complications.

Characteristic	MCDM Group: No Infection	MCDM Group: Infection	*p*
*N*	63	4	
Sex, *n* (%)			
Female	50 (79.4)	3 (75.0)	>0.999
Male	13 (20.6)	1 (25.0)	
Age, years, median (*Q1*; *Q3*)	68.00 (57.00; 76.00)	69.00 (64.25; 73.75)	0.741
ESRD cause, *n* (%)			
ADKPD	3 (4.8)	0 (0.0)	0.392
Diabetes	28 (44.4)	1 (25.0)	
Glomerulonephritis	11 (17.5)	1 (25.0)	
Hypertension	5 (7.9)	0 (0.0)	
Neoplasm	14 (22.2)	1 (25.0)	
Unknown	2 (3.2)	1 (25.0)	
Reason for removal, *n* (%)			
AVF	20 (31.7)	1 (25.0)	>0.999
AVG	2 (3.2)	0 (0.0)	
CRI	12 (19.0)	1 (25.0)	
DYS	29 (46.0)	2 (50.0)	
PD	20 (31.7)	1 (25.0)	
CTC removal site, *n* (%)			
Left femoral vein	3 (4.8)	1 (25.0)	0.095
Left jugular vein	14 (22.2)	0 (0.0)	
Left subclavian vein	1 (1.6)	0 (0.0)	
Right femoral vein	2 (3.2)	0 (0.0)	
Right jugular vein	42 (66.7)	2 (50.0)	
Right subclavian vein	1 (1.6)	1 (25.0)	
CTC time at removal, weeks, median (Q1; Q3)	57.00 (42.00; 71.50)	58.50 (43.00; 110.25)	0.741

Data presented as *n* (% of group) unless otherwise indicated. Groups compared with Fisher exact test for nominal variables and Mann–Whitney U-test for continuous variables.

**Table 5 ijerph-17-03027-t005:** Comparison of CDM patients with and without bleeding complications.

Characteristic	CDM Group: No Bleeding	CDM Group: Bleeding	*p*
*N*	70	6	
Sex, *n* (%)			
Female	46 (65.7)	3 (50.0)	0.660
Male	24 (34.3)	3 (50.0)	
Age, years, median (Q1; Q3)	65.00 (56.25; 71.25)	71.50 (63.00; 78.50)	0.244
ESRD cause, *n* (%)			
ADKPD	7 (10.0)	0 (0.0)	0.704
Diabetes	34 (48.6)	3 (50.0)	
Glomerulonephritis	7 (10.0)	1 (16.7)	
Hypertension	7 (10.0)	0 (0.0)	
Neoplasm	9 (12.9)	2 (33.3)	
Unknown	6 (8.6)	0 (0.0)	
Reason for removal, *n* (%)			
AVF	32 (45.7)	1 (16.7)	0.340
AVG	7 (10.0)	0 (0.0)	
CRI	10 (14.3)	1 (16.7)	
DYS	20 (28.6)	4 (66.7)	
PD	1 (1.4)	0 (0.0)	
CTC removal site, *n* (%)			
Left femoral vein	1 (1.4)	0 (0.0)	>0.999
Left jugular vein	21 (30.0)	2 (33.3)	
Left subclavian vein	1 (1.4)	0 (0.0)	
Right femoral vein	1 (1.4)	0 (0.0)	
Right jugular vein	45 (64.3)	4 (66.7)	
Right subclavian vein	1 (1.4)	0 (0.0)	
CTC time at removal, weeks, median (Q1; Q3)	60.00 (37.50; 87.00)	46.50 (34.50; 71.25)	0.294

Data presented as *n* (% of group) unless otherwise indicated. Groups compared with Fisher exact test for nominal variables and Mann–Whitney U-test for continuous variables.

**Table 6 ijerph-17-03027-t006:** Comparison of CDM patients with and without infection complications.

Characteristic	CDM Group: No Infection	CDM Group: Infection	*p*
*N*	73	3	
Sex, *n* (%)			
Female	46 (63.0)	3 (100.0)	0.548
Male	27 (37.0)	0 (0.0)	
Age, years, median (*Q1*; *Q3*)	65.00 (57.00; 72.00)	56.00 (46.50; 61.00)	0.182
ESRD cause, *n* (%)			
ADKPD	7 (9.6)	0 (0.0)	>0.999
Diabetes	34 (46.6)	3 (100.0)	
Glomerulonephritis	8 (11.0)	0 (0.0)	
Hypertension	7 (9.6)	0 (0.0)	
Neoplasm	11 (15.1)	0 (0.0)	
Unknown	6 (8.2)	0 (0.0)	
Reason for removal, *n* (%)			
AVF	32 (43.8)	1 (33.3)	0.690
AVG	7 (9.6)	0 (0.0)	
CRI	10 (13.7)	1 (33.3)	
DYS	23 (31.5)	1 (33.3)	
PD	1 (1.4)	0 (0.0)	
CTC removal site, *n* (%)			
Left femoral vein	1 (1.4)	0 (0.0)	>0.999
Left jugular vein	22 (30.1)	1 (33.3)	
Left subclavian vein	1 (1.4)	0 (0.0)	
Right femoral vein	1 (1.4)	0 (0.0)	
Right jugular vein	47 (64.4)	2 (66.7)	
Right subclavian vein	1 (1.4)	0 (0.0)	
CTC time at removal, weeks, median (*Q1*; *Q3*)	59.00 (36.50; 87.00)	75.00 (48.50; 84.00)	0.947

Data presented as *n* (% of group) unless otherwise indicated. Groups compared with Fisher exact test for nominal variables and Mann–Whitney U-test for continuous variables.

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
