# Peer review of "Tunnelled Haemodialysis Catheter Removal: An Underappreciated Problem, Not Always Simple and Safe"

_ijerph, 2020, doi:10.3390/ijerph17093027_

Round 1
Reviewer 1 Report
Tomasz Porazko modified the new MCDM procedure which is the simplest and
safest method of a CTC removal. The manuscript is well designed and generated so much great data to discuss. I only concern that when you discuss, could you discuss all of your results one by one.
Author Response
Author's Reply to the Review Report (Reviewer 1)
Dear Reviewer,
We would like to express our gratitude to your constructive opinions regarding our paper ( manuscript ID: ijerph-753616 ) entitled “Tunnelled haemodialysis catheter removal – underappreciated problem, not always simple and safe”. Please find below our detailed answer to your points.
Reviewer 1.
Open Review
(x) I would not like to sign my review report
( ) I would like to sign my review report
English language and style
( ) Extensive editing of English language and style required
( ) Moderate English changes required
(x) English language and style are fine/minor spell check required
( ) I don't feel qualified to judge about the English language and style
Yes Can be improved Must be improved Not applicable
Does the introduction provide sufficient background and include all relevant references?
(x) ( ) ( ) ( )
Is the research design appropriate?
(x) ( ) ( ) ( )
Are the methods adequately described?
(x) ( ) ( ) ( )
Are the results clearly presented?
( ) (x) ( ) ( )
Are the conclusions supported by the results?
(x) ( ) ( ) ( )
Comments and Suggestions for Authors
Tomasz Porazko modified the new MCDM procedure which is the simplest and
safest method of a CTC removal. The manuscript is well designed and generated so much great data to discuss. I only concern that when you discuss, could you discuss all of your results one by one.
We highly appreciate your very positive evaluation.
Sincerely yours,
Prof. Marian Klinger M.D., Ph.D. & Dr Tomasz Porażko M.D. Ph.D.
Reviewer 2 Report
This review has numerous grammatical errors that distract from the value of the report. Once appropriately corrected the manuscript will be more acceptable for publication.
Author Response
Author's Reply to the Review Report ( Reviewer 2 )
Dear Reviewer,
We would like to express our gratitude to your constructive opinions regarding our paper ( manuscript ID: ijerph-753616 ) entitled “Tunnelled haemodialysis catheter removal – underappreciated problem, not always simple and safe”. Please find below our detailed answer to your points.
Reviewer 2.
Open Review
(x) I would not like to sign my review report
( ) I would like to sign my review report
English language and style
( ) Extensive editing of English language and style required
(x) Moderate English changes required
( ) English language and style are fine/minor spell check required
( ) I don't feel qualified to judge about the English language and style
Yes Can be improved Must be improved Not applicable
Does the introduction provide sufficient background and include all relevant references?
( ) (x) ( ) ( )
Is the research design appropriate?
( ) (x) ( ) ( )
Are the methods adequately described?
( ) (x) ( ) ( )
Are the results clearly presented?
( ) (x) ( ) ( )
Are the conclusions supported by the results?
( ) (x) ( ) ( )
Comments and Suggestions for Authors
This review has numerous grammatical errors that distract from the value of the report. Once appropriately corrected the manuscript will be more acceptable for publication.
Thank you for the editorial remarks. The current version of the manuscript was corrected by native English speaker.
Sincerely yours,
Prof. Marian Klinger M.D., Ph.D. & Dr Tomasz Porażko M.D. Ph.D.
Reviewer 3 Report
A study is interesting for practicing doctors, thank you for sharing your problems and solutions.
There are two many tables in results part, some of them do not show any significant information. For example - in Table 3 there are 5 patients with bleeding complications and they are further divided into different groups according to the cause of ESRD so none of the cases in some of them, no possibility for comparison. I would offer to show just main points in tables.
A main conclusion "In our opinion MCDM is the safest and the simplest" can not be made from statistical analysis as a rate of complications was not different between two methods.
Author Response
Author's Reply to the Review Report (Reviewer 3)
Dear Reviewer,
We would like to express our gratitude to your constructive opinions regarding our paper ( manuscript ID: ijerph-753616 ) entitled “Tunnelled haemodialysis catheter removal – underappreciated problem, not always simple and safe”. Please find below our detailed answer to your points.
Reviewer 3.
Open Review
(x) I would not like to sign my review report
( ) I would like to sign my review report
English language and style
( ) Extensive editing of English language and style required
( ) Moderate English changes required
( ) English language and style are fine/minor spell check required
(x) I don't feel qualified to judge about the English language and style
Yes Can be improved Must be improved Not applicable
Does the introduction provide sufficient background and include all relevant references?
(x) ( ) ( ) ( )
Is the research design appropriate?
( ) (x) ( ) ( )
Are the methods adequately described?
(x) ( ) ( ) ( )
Are the results clearly presented?
( ) (x) ( ) ( )
Are the conclusions supported by the results?
( ) ( ) (x) ( )
Comments and Suggestions for Authors
A study is interesting for practicing doctors, thank you for sharing your problems and solutions.
There are too many tables in results part, some of them do not show any significant information. For example - in Table 3 there are 5 patients with bleeding complications and they are further divided into different groups according to the cause of ESRD so none of the cases in some of them, no possibility for comparison. I would offer to show just main points. A main conclusion "In our opinion MCDM is the safest and the simplest" cannot be made from statistical analysis as a rate of complications was not different between two methods.
Detailed comparison of patients groups, treated with studied methods, was necessary to evaluate possible influence of different factors on occurrence of a complication i.e. bleeding. The information is of a vital value for practicing colleagues as they should take special measures in certain patients cohorts. We agree that a small number of participants may result in a possible insignificance of results and it was highlighted in Discussion and conclusion chapters.
We would like to express the gratitude for such positive evaluation and respecting Reviewer constructive remark we have edited the Conclusion chapter in the more moderate and conditional mode: “ Central tunnelled catheter removal seems to be a safe and simple procedure, even at the bedside of patient. None of all previously described methods, however, were performed without complication. In our opinion MCDM offers the safer and simpler alternative way to remove CTC. The limitation of our observation is a single center experience, retrospective character and a small sample of patients. Even so, our data may contribute to improve of CTC removal technique. The presented modified cut down method may increase the safety of a central tunnelled catheter removal and protect against nonintentional catheter migration into central veins or enhance a stuck catheter removal.”
Sincerely yours,
Prof. Marian Klinger M.D., Ph.D. & Dr Tomasz Porażko M.D. Ph.D.
Reviewer 4 Report
Poorly written paper that would be better presented as a case report and review of literature. With suggested modifications to reduce complications. The scientific approach to presentation does not make this paper any stronger due to retrospective nature and population size as acknowledged. Is this the correct journal for exposure to the the targeted audience?
Tunnelled haemodialysis catheter removal – underappreciated problem, not always simple and safe.
Tomasz Porazko, Jacek Hobot, Zbigniew Ziembik and Marian Klinger.
Department of Nephrology and Internal Medicine, Institute of Medical Sciences, University of Opole, Opole, Poland
Page 1, Line 13: “Optimal care of patients treated with tunneled central catheter (CTC ) ……” Suggest consistency with text and abbreviations ‘Optimal care of patients treated with central tunneled catheter ( CTC )…’
Page 1, Line 28: “….significant number of prevailed end-stage kidney disease (ESKD) patients.” Suggest eliminate prevailed, it is out of context. Also suggest using standard nomenclature for ESRD. ‘…significant number of end-stage renal disease (ESRD) patients.’
Page 1, Line 30: “Comparing to that, a negligible interest is paid to a CTC removal…..” Suggest ‘ In comparison to CTC insertion, negligible interest is paid to CTC removal….’
Page 1, Line 31: “…which is believed to be simple and safe event in outpatient suite.” Suggest ‘….which is believed to be simple and safe procedure in an outpatient setting.’
Page 1, Line 32: “However, there are some significant risks that cannot be omitted.” Suggest ‘However, there are some significant risks that cannot be overlooked.’
Page 1, Line 32: “There are two main indications for the operation.” Suggest ‘There are two main indications for CTC removal.’
Page 1, Line 33: “First, scheduled, when a CTC is not needed……” Suggest ‘First, scheduled removal, when a CTC is no longer needed….’
Page 1, Line 35: “Second, urgent, due to unexpected complications like catheter……” Suggest ‘Second, emergent removal, due to unexpected complications like catheter…..’
Page 1, Line 36: “…….thrombotic dysfunction requiring exchange.” Suggest ‘……thrombotic dysfunction requiring exchange or resighting.”
Page1, Line 38: “Despite its simplicity and safety both methods can be seriously complicated.” Suggest ‘Despite their simplicity and safety both methods can potentially lead to significant complications.’
Page1, Line 39: “….case of technique failure of CDM, there were reports of a intravascular part of CTC migration of into….” Suggest ‘…..case of technique failure of CDM, there are reports of an intravascular part of CTC migration into the …..’
Page 1, Line 40: “From 2008 till 2015 the CDM was regularly used,….” Suggest ‘From 2008 till 2015 the CDM technique was regularly used,…’
Page 1, Line 41: “However after observation of two patients with a CTC part migration….” Suggest ‘However, after observation of two patients with a CTC segment migration….’
Page1, Line 42: “…and one patient with stuck CTC, the own modification of catheter removal was elaborated….” Suggest ‘….and one patient with retained CTC, our own modification to CTC removal was devised…..”
Page1, Line 43: “In principle, modified approach, after a CTC cuff was released, an 44 intravascular part was pulled out and cut down just after instead of before, as it was done in previous,….” Suggest ‘In our modified approach, after the CTC cuff was released, the intravascular part was pulled out and then cut down, instead of being cut down prior, as it was done in our unmodified technique of CDM.’
Page2, Line 45: “It prevents inadvertent complications i.e. migration of distal part of CTC into SVC or RA, as might occur as it was seen in our practice and also reported by others.” Suggest ‘We believed (hypothesized) that this would prevent the inadvertent complications of migration of distal part of CTC into SVC or RA or retained segment, as previously observed in our practice and also reported by others.’
Page 2, Line 47: “This report presents the experience in application of the new variant modified cut down 48 method (MCDM ) comparing with previously used, standard CDM.” Suggest “We report our retrospective analysis of our experience in application of the new variant modified cut down technique ( MCDM ) in comparison to the standard CDM technique.’
Author Response
Dear Reviewer,
We would like to express our gratitude to your constructive opinions regarding our paper ( manuscript ID: ijerph-753616 ) entitled “Tunnelled haemodialysis catheter removal – underappreciated problem, not always simple and safe”. Please find below our detailed answer to your points.
Open Review
(x) I would not like to sign my review report
( ) I would like to sign my review report
English language and style
(x) Extensive editing of English language and style required
( ) Moderate English changes required
( ) English language and style are fine/minor spell check required
( ) I don't feel qualified to judge about the English language and style
Yes Can be improved Must be improved Not applicable
Does the introduction provide sufficient background and include all relevant references?
( ) ( ) (x) ( )
Is the research design appropriate?
( ) ( ) (x) ( )
Are the methods adequately described?
( ) ( ) (x) ( )
Are the results clearly presented?
( ) ( ) (x) ( )
Are the conclusions supported by the results?
( ) ( ) (x) ( )
Comments and Suggestions for Authors
Poorly written paper that would be better presented as a case report and review of literature. With suggested modifications to reduce complications. The scientific approach to presentation does not make this paper any stronger due to retrospective nature and population size as acknowledged. Is this the correct journal for exposure to the targeted audience?
We obviously respect the Reviewer #4 right to strong criticism, thank you. However, in fact the essential remarks are minor, and the evaluations by three remaining Reviewers are profoundly more favourable. Thank you for linguistic suggestions which were taken into consideration. The manuscript was, as per kind suggestion, checked by native English speaker. Concerning the objection whether International Journal of Environmental Research and Public Health is the appropriate place for publication we would like to inform that senior author, active journal reviewers board member in field of dialysis problems. Moreover we have chosen the journal in response to kind invitation for paper publication. Furthermore, it is common practice to search for literature by key words, and journal name has no direct significance.
Tunnelled haemodialysis catheter removal – underappreciated problem, not always simple and safe.
Tomasz Porazko, Jacek Hobot, Zbigniew Ziembik and Marian Klinger.
Department of Nephrology and Internal Medicine, Institute of Medical Sciences, University of Opole, Opole, Poland
Page 1, Line 13: “Optimal care of patients treated with tunnelled central catheter (CTC ) ……” Suggest consistency with text and abbreviations ‘Optimal care of patients treated with central tunnelled catheter ( CTC )…’
Suggestion introduced. Page 1, Line 13: “Optimal care of patients treated with central tunnelled catheter ( CTC )…”
Page 1, Line 28: “….significant number of prevailed end-stage kidney disease (ESKD) patients.” Suggest eliminate prevailed, it is out of context. Also suggest using standard nomenclature for ESRD. ‘…significant number of end-stage renal disease (ESRD) patients.’
Suggestion introduced. Page 1, Line 28: Word “prevailed “ was eliminated as well as end - stage renal disease ( ESRD ) nomenclature was introduced ( page 1, line 30; page 4, line 96 ; page 7, line 182; Tables 1, 3, 4, 5, 6 ).
Page 1, Line 30: “Comparing to that, a negligible interest is paid to a CTC removal…..” Suggest ‘ In comparison to CTC insertion, negligible interest is paid to CTC removal….’
Suggestion was taken into consideration and Introduction chapter was edited by English language editor as presented below. Page 1 to 2, line 29 to 50 : “ Central tunneled catheters (CTC) are widely used as a vascular access for hemodialysis in a significant number of prevailed end-stage renal disease (ESRD) patients [1]. The main attention is focused on the improvement of the line insertion technique and everyday care to maintain a high level of CTC patency without thrombotic dysfunction or infective complications. In comparison with that, a negligible interest is paid to CTC removal, which is believed to be a simple and safe even in the outpatient setting [2, 3]. However, there are some significant risks that cannot be ignored [2, 3, 4]. There are two main indications for the operation. The first one is a scheduled removal, when a CTC is not needed anymore, i.e. for a patient with matured AV fistula (AVF) or graft (AVG), as well as with a successfully functioning kidney transplant. The second indication results from medical emergency due to unexpected complications, like catheter related infection (CRI) or thrombotic dysfunction requiring exchange. Generally, two techniques of CTC removal are widely used: the cut-down method (CDM) [5, 6, 7] and the traction method (TM) [2, 3], or each one with a few modifications [7, 8]. Despite their simplicity and safety, both methods can be seriously complicated. In the case of the technique failure of CDM, there were the reports of an intravascular part of CTC migrating into superior vena cava (SVC) or right atrium (RA) [4]. CDM was regularly used both in our center and in its satellite units from 2008 till 2015. However, after the observation of two patients with a CTC part migration and one patient with a stuck CTC, our own modification of catheter removal was devised in 2015 and it has been introduced ever since. In principle, in the modified approach, after the CTC cuff was released, the intravascular part was pulled out and cut down just after – instead of before, as it was done previously in the CDM method “
Page 1, Line 31: “…which is believed to be simple and safe event in outpatient suite.” Suggest ‘….which is believed to be simple and safe procedure in an outpatient setting.’
Suggestion was taken into consideration and Introduction chapter was edited by English language editor as presented above ( Page 1 to 2, line 29 to 50 ).
Page 1, Line 32: “However, there are some significant risks that cannot be omitted.” Suggest ‘However, there are some significant risks that cannot be overlooked.’
Suggestion introduced. Suggestion was taken into consideration and Introduction chapter was edited by English language editor as presented above ( Page 1 to 2, line 29 to 50 ).
Page 1, Line 32: “There are two main indications for the operation.” Suggest ‘There are two main indications for CTC removal.’
Suggestion introduced. Suggestion was taken into consideration and Introduction chapter was edited by English language editor as presented above ( Page 1 to 2, line 29 to 50 ).
Page 1, Line 33: “First, scheduled, when a CTC is not needed……” Suggest ‘First, scheduled removal, when a CTC is no longer needed….’
Suggestion introduced. Suggestion was taken into consideration and Introduction chapter was edited by English language editor as presented above ( Page 1 to 2, line 29 to 50 ).
Page 1, Line 35: “Second, urgent, due to unexpected complications like catheter……” Suggest ‘Second, emergent removal, due to unexpected complications like catheter…..’
Suggestion introduced. Suggestion was taken into consideration and Introduction chapter was edited by English language editor as presented below ( Page 1 to 2, line 29 to 50 ).
Page 1, Line 36: “…….thrombotic dysfunction requiring exchange.” Suggest ‘……thrombotic dysfunction requiring exchange or resighting.”
Suggestion not introduced. Resighting in the meaning: revision, inspection, second look, to see again is out of context considering catheter removal.
Page1, Line 38: “Despite its simplicity and safety both methods can be seriously complicated.” Suggest ‘Despite their simplicity and safety both methods can potentially lead to significant complications.’
Suggestion was taken into consideration and Introduction chapter was edited by English language editor as presented above ( Page 1 to 2, line 29 to 50 ).
Page1, Line 39: “….case of technique failure of CDM, there were reports of a intravascular part of CTC migration of into….” Suggest ‘…..case of technique failure of CDM, there are reports of an intravascular part of CTC migration into the …..’
Suggestion was taken into consideration and Introduction chapter was edited by English language editor as presented above ( Page 1 to 2, line 29 to 50 ).
Page 1, Line 40: “From 2008 till 2015 the CDM was regularly used,….” Suggest ‘From 2008 till 2015 the CDM technique was regularly used,…’
Suggestion was taken into consideration and Introduction chapter was edited by English language editor as presented above ( Page 1 to 2, line 29 to 50 ).
Page 1, Line 41: “However after observation of two patients with a CTC part migration….” Suggest ‘However, after observation of two patients with a CTC segment migration….’
Suggestion was taken into consideration and Introduction chapter was edited by English language editor as presented above ( Page 1 to 2, line 29 to 50 ).
Page1, Line 42: “…and one patient with stuck CTC, the own modification of catheter removal was elaborated….” Suggest ‘….and one patient with retained CTC, our own modification to CTC removal was devised…..”
Suggestion was taken into consideration and Introduction chapter was edited by English language editor as presented above ( Page 1 to 2, line 29 to 50 ).
Page1, Line 43: “In principle, modified approach, after a CTC cuff was released, an 44 intravascular part was pulled out and cut down just after instead of before, as it was done in previous,….” Suggest ‘In our modified approach, after the CTC cuff was released, the intravascular part was pulled out and then cut down, instead of being cut down prior, as it was done in our unmodified technique of CDM.’
Suggestion was taken into consideration and Introduction chapter was edited by English language editor as presented above ( Page 1 do 2, line 29 to 50 ).
Page2, Line 45: “It prevents inadvertent complications i.e. migration of distal part of CTC into SVC or RA, as might occur as it was seen in our practice and also reported by others.” Suggest ‘We believed (hypothesized) that this would prevent the inadvertent complications of migration of distal part of CTC into SVC or RA or retained segment, as previously observed in our practice and also reported by others.’
Suggestion introduced. Page 2, line 50 to 54 : “ We believed (hypothesized) that this would prevent the inadvertent complications of migration of distal part of CTC into SVC or RA or retained segment, as previously observed in our practice and also reported by others.”
Page 2, Line 47: “This report presents the experience in application of the new variant modified cut down 48 method (MCDM ) comparing with previously used, standard CDM.” Suggest “We report our retrospective analysis of our experience in application of the new variant modified cut down technique ( MCDM ) in comparison to the standard CDM technique.’
Suggestion introduced. Page 2 line 55 to 58 : “We report our retrospective analysis of our experience in application of the new variant modified cut down technique ( MCDM ) in comparison to the standard CDM technique.”
Sincerely yours,
Prof. Marian Klinger M.D., Ph.D. & Dr Tomasz Porażko M.D. Ph.D.